# pH-Responsive Nanocarriers in Cancer Therapy

**DOI:** 10.3390/polym14050936

**Published:** 2022-02-26

**Authors:** Nour M. AlSawaftah, Nahid S. Awad, William G. Pitt, Ghaleb A. Husseini

**Affiliations:** 1Department of Chemical Engineering, College of Engineering, American University of Sharjah, Sharjah P.O. Box. 26666, United Arab Emirates; g00051790@alumni.aus.edu (N.M.A.); nawad@aus.edu (N.S.A.); 2Materials Science and Engineering Program, College of Arts and Sciences, American University of Sharjah, Sharjah P.O. Box. 26666, United Arab Emirates; 3Chemical Engineering Department, Brigham Young University, Provo, UT 84602, USA; pitt@byu.edu

**Keywords:** nanoparticles, pH, drug delivery, cancer

## Abstract

A number of promising nano-sized particles (nanoparticles) have been developed to conquer the limitations of conventional chemotherapy. One of the most promising methods is stimuli-responsive nanoparticles because they enable the safe delivery of the drugs while controlling their release at the tumor sites. Different intrinsic and extrinsic stimuli can be used to trigger drug release such as temperature, redox, ultrasound, magnetic field, and pH. The intracellular pH of solid tumors is maintained below the extracellular pH. Thus, pH-sensitive nanoparticles are highly efficient in delivering drugs to tumors compared to conventional nanoparticles. This review provides a survey of the different strategies used to develop pH-sensitive nanoparticles used in cancer therapy.

## 1. Introduction

Globally, cancer kills a staggering 9.3 million people annually [1]. Conventional methods to treat cancer, such as chemotherapy, are associated with severe and often debilitating systemic side effects [2]. Therefore, the encapsulation of anti-cancer drugs in nano-sized carrier systems has been proposed as an approach to increase the drug(s) concentration at a localized site while reducing their detrimental side effects. Nanoparticles (NPs) are small in size (1–500 nm), which endows them with unique properties [3,4,5]. Figure 1 shows the main advantages of using NPs in drug delivery.

A wide variety of materials has been used to synthesize NPs used to deliver drugs. Generally, NPs can be prepared from either organic or inorganic materials. The preparation of inorganic NPs usually involves elemental metals, metal oxides and metal salts. Examples of inorganic NPs include quantum dots (QDs), gold NPs, silica NPs, and magnetic NPs [6,7]. On the other hand, organic NPs are composed of natural or synthetic organic molecules such as polymeric-based and lipid-based NPs. Polymeric NPs include polymersomes, dendrimers, nanospheres, hydrogels, and polymeric micelles, while lipidic NPs include liposomes, solid lipid nanoparticles (SLNs), and nano-emulsions [7,8,9].

The physical and chemical characteristics of NPs, such as size and shape, can significantly affect their behavior inside the body. It is anticipated that successful NPs can achieve long circulation time, in the blood, to ensure efficient delivery of the encapsulated drugs to their targets. However, the immune system often recognizes these NPs as foreign substances and works to clear them from the body before they are able to reach their targeted site. Studies have reported that the kidneys excrete NPs (into the urine) with diameters less than 6 nm. In contrast, much larger NPs with diameters between 100–7000 nm are recognized and cleared by the organs of the reticuloendothelial system (RES). However, NPs that have very small diameters (less than 100 nm) can fall into the fenestrae between the cells that make up the endothelial cell lining of the blood vessels and, thus, will not be detected nor eliminated from the body [10]. The advantages and disadvantages of various organic and inorganic NPs are briefly summarized in Table 1 and Table 2.

As mentioned earlier, NPs can be designed to target specific body locations. They can deliver their payloads either through passive or active targeting. Passive targeting can be best defined as the accumulation of NPs in or beyond the fenestrae of tumor vessels, whose defining characteristic is having a disordered and leaky vasculature. This is referred to as the enhanced permeability and retention (EPR) effect [18]. The selectivity of NPs towards their targeted sites can be enhanced by using targeting moieties that can be conjugated to the NPs; this approach is referred to as active targeting [19,20,21,22].

Controlled drug delivery systems enable the spatiotemporal control of drug release, i.e., delivering the encapsulated drugs to the targeted site and releasing it at a rate that provides the desired concentration. Drug release from the different NPs can be controlled using selective triggering mechanisms [23,24]. Stimulus-responsive NPs are designed to maintain their structure while circulating in the body and release their payload upon exposure to one or more of the stimuli mentioned above [25,26]. These triggers can be internal (local temperature, pH, redox, and enzymes), or external (applied heating, ultrasound [US], magnetic field, and light) [19,27]. Table 3 summarizes the advantages and disadvantages of the aforementioned triggering mechanisms. Currently, research efforts are directed towards combining internal and external triggers to improve release efficiency [28]. This review explores the advances made in pH-responsive NPs and their applications in treating cancer.

## 2. pH-Sensitive Biomaterials and Particles

Tumors have a unique microenvironment characterized by elevated temperatures, elevated expression of certain enzymes, a redox potential biased toward reduction, and acidic pH (~6.5). The low extracellular pH of solid tumors is due to the preference of tumor tissues to undergo anaerobic respiration [33,34,35,36]. Accordingly, pH-responsive NPs have been extensively researched to deliver drugs to tumors. Those NPs release their payload in response to changes in acidic conditions [37]. To achieve this, two different mechanisms can be applied by incorporating protonatable groups or forming acid-labile bonds [38,39].

pH-triggered protonation/ionization is widely used to produce pH-responsive NPs. A number of ionizable groups are incorporated into the NPs structure. The exposure to low pH causes the protonation or charge reversal of the incorporated functional groups, thus, disturbing the hydrophilic-hydrophobic equilibrium inside the NP, leading to the disassembly of the nanocarrier’s structure and subsequent release of the encapsulated payload. Amino, carboxyl, sulfonate, and imidazolyl groups are among the most used ionizable groups [38,39,40]. Drug release from such NPs can occur through three mechanisms: precipitation, aggregation, or dissociation depending on the acid dissociation constant (pK_a_) of the introduced functional group [38,41].

Based on the mechanism of protonation/ionization, pH-sensitive polymers are divided into anionic and cationic, which is determined by their charge at physiological pH. Cationic polymers change from being non-protonated (unionized/hydrophobic) to deprotonated (ionized/hydrophilic) with the drop in pH, whereas polymers that are anionic change from being hydrophilic to hydrophobic when the pH decreases below their pK_a_ [42]. This change in hydrophilicity leads to the reformation of the polymeric nanocarrier system composed of these polymers and produces the subsequent release of the drug. Examples of cationic and anionic polymers and details of their conformational changes in response to the change in pH are detailed in Table 4.

With regard to lipidic NPs, Dioleoylphosphatidylethanolamine (DOPE) is a widely used lipidic pH-sensitive NPs. At physiological pH, DOPE has an inverted cone shape due to the presence of intermolecular forces between the polar head groups and the amine group, giving it a reverse hexagonal (H_II_) shape. To form lipid bilayer vesicles, a lipid with a larger head group, such as cholesteryl hemisuccinate (CHEMS), must become incorporated. When the pH is low, a change in the conformation of the carboxylic group of CHEMS from a cone-shaped to a cylindrical-shaped occurs as it becomes protonated. This will result in vesicular destabilization [38,43]. In addition to the physical changes, pH-triggered protonation/ionization can also cause chemical changes. A drop in pH can cleave the covalent acid-labile bonds on the surface or within the NPs. The most common pH-sensitive chemical bonds are imine, hydrazone, oxime, amide, ethers, orthoesters, acetals, and ketals [39]. An important mechanistic example is acid-labile bond cleavage for poly(ethylene glycol) (PEG) detachment. This chemistry has been developed because PEGylation is used to make the NPs more stable with better circulation time; however, PEGylated NPs suffer from low uptake by the cells and the subsequent drug release inside the cells, a phenomenon known as the “PEG dilemma.” To solve this problem, pH-sensitive PEG detachment, where the PEG shell detaches from the NP at the tumor site or in the endosome due to the change in pH, is employed [38,44]. Table 5 below provides a summary of acid-labile chemical bonds, while Figure 2 depicts the pH-triggered release from NPs, which include the use of protonable/charge shifting groups, cleavage of acid-labile bonds, or the use of crosslinkers which combines charge shifting polymers with either non-cleavable bonds, leading to the swelling of NPs, or acid-labile bonds which lead to pH-triggered disassembly of the NPs [42].

## 3. pH-Responsive Nanocarriers for Cancer Therapy

Cancer cells are characterized by increased glucose uptake to sustain their rapid proliferation, and sometimes poor vasculature to adequately supply oxygen; accordingly, cancer cells often are biased toward the anaerobic path for glucose metabolism, which produces lactic acid as a byproduct of incomplete oxidation [36]. The increased levels of lactic acid decrease the pH of the tumor environment; this is referred to as the “Warburg effect”. Therefore, in cancer therapy, low-pH-responsive NPs will release the encapsulated chemotherapeutic agents upon encountering the acidic tumor microenvironment. Several research groups have worked on endowing different NPs with pH sensitivity. The following sections will detail some of these experimental observations.

### 3.1. pH-Responsive Metal-Organic Frameworks (MOFs)

Metal–organic frameworks (MOFs) are hybrid, i.e., organic and inorganic, porous crystalline materials composed of metal ions and/or clusters connected by organic linkers [46,47]. MOFs have properties that make them quite effective as drug delivery systems, including high surface area and high porosity (which increase drug loading efficiency), open metal sites for physical and chemical interactions, and ease of functionalization. Drugs can be loaded into MOFs or attached to their surfaces through various inter- and intra-molecular bonds, e.g., hydrogen and covalent bonds, van der Waals forces and electrostatic interconnections [46,47,48]. Various methods can be used to synthesize MOFs, which are reviewed in detail in [47] and summarized in Figure 3 below.

pH-sensitive MOFs are widely investigated because their bonds’ arrangement is sensitive to environmental pH [46]. Several research groups have worked on developing pH-responsive MOFs. Duan et al. [49] prepared a pH-responsive MOF-based NP for the co-delivery of drugs. The MOF contained immunostimulatory unmethylated cytosine-phosphate-guanine oligonucleotide (CpG) and tumor-associated antigens (TAAs) for cancer immunotherapy. Antigen release reached around 60% when exposed to pH 5.0. Moreover, the developed system showed enhanced antitumor activity when employed in vivo against B16-OVA melanoma cancers. In another study, Pandey et al. [50] synthesized a hyaluronic acid (HA) coated MOF delivery system. Titanocene was loaded into a lactoferrin (Lf) protein matrix, which was then enclosed, along with 5-FU, in a ZIF-8 MOF coated with a Lenalidomide-HA conjugate linked via a hydrazone linkage (LND-HA@ZIF-8@Lf-TC). In vitro experiments were conducted using U87MG glioblastoma cells. The developed system showed pH sensitivity and enhanced anti-cancer activity through the disruption of intracellular IL-6 and TNFα levels. Release of 5-FU from LND-HA@ZIF-8@Lf-TC following 48 h of incubation at pH 5.5 was 92.59 ± 3.5%, while at pH 7.4, it was 18.30 ± 2.7%. The decrease in cell viability was 44.2 ± 3.7% and 58.8 ± 3.3% after 24 h and 48 h, respectively.

### 3.2. pH-Responsive Gold Nanoparticles

Gold nanoparticles (Au NPs) are another interesting type of NPS and have received significant attention because of their high surface area, increased loading, and simplicity of functionalization with thiolated molecules. Several research groups have investigated pH-sensitive Au NPs. For example, Kumar et al. [51] developed doxorubicin (DOX)-loaded pH-responsive Au NPs decorated with the short tripeptide Lys–Phe–Gly (KFG). The developed NPs were tested using cervical carcinoma (HeLa) cells, human embryonic kidney transformed (HEK 293 T), and glioblastoma (U251) cell cultures. The MTT assay showed that a lower number of viable cells was recorded in the cells incubated with DOX-loaded KFG-Au NPs compared to the free DOX. The flow cytometry results showed greater internalization of the DOX-KFG-Au NPs than free DOX in HeLa cells. In vivo testing was conducted in breast cancer (BT-474) cell xenograft nude mice, which showed that DOX-KFG-AuNP treatment groups had significantly smaller tumor volumes than those treated with free DOX.

Samadian et al. [52] designed a PEG and folic acid (FA)-functionalized graphene oxide (GO) decorated with Au NPs (GO–PEG–FA/GN). The developed hybrid system encapsulated DOX, and its anti-cancer efficacy was tested using human breast cancer (MCF-7) cells. With regard to pH-responsiveness, GO-PEG-FA/GNs showed higher drug release at pH 4.0 compared to that measured at a pH of 7.4, which was attributed to the weakening of the π-π stacking and hydrophobic interactions between the drug molecules and the NPs. Furthermore, GO–PEG–FA/GNPs were more toxic to the cancer cells compared to the free drug. In another study, Khodashenas et al. [53] investigated methotrexate (MTX) drug delivery in breast cancer treatment. MTX was loaded into gelatin-coated spherical (50 and 100 nm in diameter) NPs and nanorod-shaped (Au NRs, 20, 50, and 100 nm in length) NPs. The characterization findings showed that the entrapment efficiency of the spherical AuNPs was higher than that of Au-nanorods (Au-NRs). However, the highest release rate of MTX was achieved using gelatin-coated Au-NRs at pH 5.4 (40 °C). Moreover, the highest cytotoxicity was recorded when MTX loaded gelatin-coated Au-NRs were used.

### 3.3. pH-Responsive Dendrimers

Dendrimers are very ordered, branched polymeric nanostructures containing a core from which symmetric branches (dendrons) grow radially outward. Drugs and other molecules can be incorporated into dendrimers through encapsulation, conjugation, or complexation [54]. Their hyperbranched architecture and high loading capacity make dendrimers attractive drug delivery vehicles. Karimi and Namazi [55] covalently attached a triazine dendrimer to a magnetic carbon NP using a maltose molecule (Fe_3_O_4_@C@TD-G3). This system was then allowed to react with graphene QDs to form the final structure (Fe_3_O_4_@C@TDGQDs). The developed nanocarrier was loaded with DOX and its antitumor activity was assessed against human lung cancer (A549) cells at pH 5, 6.8, and 7.4. The in vitro DOX release from Fe_3_O_4_@C@TDGQDs was tested using phosphate buffer saline (PBS) at the aforementioned pH values at time intervals at 37 °C. The results showed that DOX release a pH-dependent process. The toxicity results indicated that free DOX was less toxic than that delivered from Fe_3_O_4_@C@TDGQDs at all the tested concentrations.

Zhang et al. [56] studied the imaging-guided anti-cancer ability of a dendrimer and aptamer grafted persistent luminescent nanoprobe. Polyamide-amine (PAMAM) dendrimer-grafted persistent luminescence nanoparticles (PLNPs) were functionalized with aptamer AS1411 and loaded with DOX. The results showed that a stronger luminescence signal was detected from the PLNPs-PAMAM-AS1411/DOX in the HeLa cells in comparison to normal cells, indicating an increased uptake. The DOX release from PLNPs-PAMAM-AS1411/DOX at pH 5.0 was around 60%, compared to a 10% release at physiological following 36 h of incubation. Finally, a luminescence signal was recorded inside the tumor tissues of the mice treated with PLNPs-PAMAM-AS1411/DOX, while the PLNPs-PAMAM-DOX group did not give a signal, suggesting that the functionalization with AS1411 aptamer achieved active targeting effects and promoted the accumulation of the nanoprobes at the tumor site.

### 3.4. pH-Responsive Polymeric Micelles

Polymeric micelles (PMs) are self-assembling colloidal NPs with a hydrophilic shell and a hydrophobic core ranging in size between 10 and 200 nm. A wide variety of polymers can be used to make PMs, including amphiphilic di-block copolymers (e.g., polystyrene and PEG), triblock copolymers (e.g., poloxamers), graft (e.g., G-chitosan), and ionic (e.g., poly(ethylene glycol)-poly(ε-caprolactone)-g-polyethyleneimine) copolymers. The amphiphilic block copolymers making up PMs self-assemble above a given concentration referred to as the critical micellar concentration (CMC). In diluted aqueous solutions, the polymers exist separately as unimers and act as surfactants to reduce interfacial tension. When the bulk solution saturation concentration exceeds the CMC, these unimers aggregate to form PMs. This makes the CMC the most important parameter for controlling the thermodynamic stability of PMs [16,27,57]. Drugs can be covalently conjugated to the polymers making up the PMs or physically loaded into PMs. Depending on the method of PM preparation and on the properties of the drug(s), these therapeutics can be encapsulated into PMs during their formation or incorporated post-formation. Commonly used preparation methods for PMs include direct dissolution, dialysis, emulsion with solvent evaporation, and solution-casting followed by film hydration. Generally, lipophilic drugs are hosted in the hydrophobic core of the PMs, while hydrophilic drugs are located in the shell (refer to Figure 4) [27,57,58].

Furthermore, PMs with specific functional groups responsive to endogenous stimuli such as pH, redox, and enzymes have been studied extensively for the controlled delivery of therapeutics at specifically targeted sites, particularly in cancer therapy. For instance, Domiński et al. [59] synthesized triblock copolymer poly(ethylene glycol)-b-polycarbonate-b-oligo([R]-3-hydroxybutyrate) (PEG-PKPC-oPHB) PMs encapsulating DOX and 8-hydroxyquinoline glucose (8HQ-glu)- and galactose conjugates (8HQ-gal). Drug release from this system was triggered by increasing the hydrophilicity of the originally hydrophobic core through acid-triggered hydrolysis of the ketal groups. In vitro release experiments showed that drug release increased significantly at lower pH (46% at pH 7.4 and 77% at pH 5.5). The MTT assay results showed that the loaded micelles had improved anti-cancer efficacy. Furthermore, drug glyco-conjugation and pH-responsive PMs showed synergistic effects, which significantly increased their ability to inhibit the proliferation of cancer cells.

Hsu et al. [60] used amphiphilic chitosan-g-mPEG/DBA conjugates to form PMs sequestering indocyanine green dye (ICG). Characterization tests showed that the synthesized PMs had a hydrophobic hybrid chitosan/DBA core and a hydrophilic PEG shell. In vitro IGC (a model drug) release experiments were conducted using the dialysis method, and the results showed that the cumulative drug release at pH 5.0 (23%) was higher than that at the physiological pH of 7.4 (9%). The pH-induced release was due to the cleavage of benzoic-imine bonds between chitosan and DBA. An MTT assay was performed to determine the cytotoxicity of the developed system, in which MCF-7 cells were treated with high concentration (29–394 μg/mL) ICG-encapsulating PMs for 24 h. The results showed high cell viability (87%), indicating the non-toxic nature of these PMs. Despite their advantages as drug delivery carriers, PMs suffer from stability issues that hinder their clinical translation. The instability of PMs stems from the dynamic shift of polymer chains between the micellar and bulk phases. Proposed solutions include utilizing non-covalent interactions (e.g., hydrophobic, electrostatic, hydrogen bonding, and coordination interactions) to improve the stability of PMs. Of these, hydrophobic interactions were found to strongly influence the stability and loading efficiency of micelles. Accordingly, Son et al. [61] synthesized PMs using block copolymers, poly(ethylene glycol)-blockpoly(cyclohexyloxy ethyl glycidyl ether)s (mPEG-b-PCHGE) with an acetal group as the pH-cleavable linkage. In vitro release results showed higher stability and better pH responsiveness due to the addition of the hydrophobic CHGE block. Table 6 summarizes some studies relevant to pH-responsive micelles in cancer therapy, and Table 7 lists some polymeric micelles-based drugs used for cancer therapy.

### 3.5. pH-Responsive Liposomes

Liposomes are spherical vesicles consisting of amphiphilic phospholipids arranged in concentric bilayers around an aqueous core. The hydrophobic tails of the phospholipids are directed toward the interior of the bilayer, while the hydrophilic heads are directed towards the aqueous environment (refer to Figure 5). The structure of liposomes offers them the unique ability to encapsulate both hydrophilic (in core) and hydrophobic drugs (in bilayer). Liposomes are considered one of the most successful DDSs because of their biocompatibility, biodegradability, and non-toxic and nonimmunogenic nature. The amphiphilic nature of phospholipids not only grants them the ability to encapsulate hydrophilic and hydrophobic drugs, but also enables them to mimic natural cell membranes promoting efficient cellular uptake [66,67]. Moreover, the surfaces of liposomes can be easily functionalized with stealth-imparting polymers (e.g., PEG) and/or other targeting moieties. PEGylating the liposomes improve their circulation time in the body by hindering their interactions with the organs of the RES. Table 8 details some clinically approved liposomes-based products for cancer therapy.

The main limitation of conventional and stealth liposomes is that they cannot be delivered directly to specific target cells; this gave rise to the development of ligand-targeted liposomes. These liposomes are decorated with one or more ligands that can target specific receptors overexpressed on the surfaces of a specific cell type, such as cancer cells, thus, increasing the liposome’s selective therapeutic efficiency. In addition, drug release from liposomes can be controlled using internal or external stimuli [19,27,69]. Among internal stimuli-sensitive liposomes, pH-responsive liposomes are quite popular in cancer therapy because they respond to the acidic nature of the tumor microenvironment to release their contents. pH-responsive liposomes usually consist of a neutral lipid such as a phosphatidylamine derivative, and a weakly acidic amphiphile, such as CHEMS. In the acidic tumor environment, the negatively charged phospholipid destabilizes, leading to better fusion with the cellular and/or endosomal membrane and the subsequent release of liposomal contents [70,71].

Many studies have focused on developing pH-sensitive liposomes for cancer therapy; for instance, Zhai et al. [72] synthesized pH-responsive DOX-liposomes using the acid-sensitive peptide DVar7 (DOPE-DVar7-lip@DOX). The anti-cancer activity of DOPE-DVar7-lip@DOX was investigated in vitro and in vivo using flow cytometry and near-infrared (NIR) fluorescent imaging. The DOX release from DOPE-DVar7-lip@DOX at pH 5.3 was nearly five times more than DOX release at pH 7.4. The in vitro uptake was evaluated in cervical carcinoma (HeLa) and breast cancer (MDA-MB-435S) cells, and the flow cytometry results for DOPE-DVar7-liposomes showed increased uptake in tumor cells (MDA-MB-435S: 7.55 ± 0.04 at pH 5.3 vs. 6.97 ± 0.01 at pH 7.4, *p* < 0.001; HeLa: 7.75 ± 0.03 at pH 5.3 vs. 7.40 ± 0.02 at pH 7.4, *p* < 0.001). The in vivo therapeutic efficiency of the developed liposomal system was evaluated in mice inoculated with MDA-MB-435S cells. The group treated with DOPE-DVar7-lip@DOX showed the best therapeutic efficacy with tumor volumes of 86.73 ± 6.51 mm^3^ compared to 196.10 ± 17.06 mm^3^ for the free DOX.

In another study, Zarrabi et al. [73] attached citraconic anhydride (CA) to PEG and DSPE to impart their curcumin-loaded liposomes with pH sensitivity. Improved curcumin release was observed at pH 6.6, with the release profile showing burst release kinetics during the first 24 h followed by sustained release. This release pattern was attributed to the disruption of the pH-responsive bond and the subsequent release of the CA-PEG layer, which, in turn, released the curcumin trapped in the polymeric shell as well as some of the curcumin contained inside the liposome. In the sustained release stage, curcumin inside the liposomes was released in response to the change in pH.

Wang et al. [74] synthesized a novel zwitterionic lipid 2-(4-((1,5-bis(octadecyloxy)-1,5-dioxopentan-2-yl) carbamoyl) pyridin-1-ium-1-yl) acetate (DCPA) and used water as the pH-responsive functional group. The DCPA-H_2_O liposomes were loaded with the red-fluorescent rhodamine dye as a model drug. Specific accumulation of the DCPA-H_2_O liposomes at the acidic tumor site became evident after 6 h from the time of injection and was 11-times higher than whole-body distribution. Additional studies are detailed in Table 9.

## 4. Challenges and Opportunities

pH-responsive NPs have versatile chemical structures that allow them to include different pH-responsive groups or bonds to modulate drug release under the acidic conditions of tumors. Generally, the encapsulation of chemotherapeutic drugs inside pH-responsive NPs is a plausible method to lengthen the circulation time of the encapsulated drugs and their retention inside the NPs in physiological pH. pH-responsive NPs are also able to improve the pharmacokinetics and biodistribution of the drugs. This is essential for delaying the metabolism and the subsequent clearance of drugs. Furthermore, pH-responsive NPs also allow a controlled release of the encapsulated drug at acidic pH upon reaching the desired site. Despite their promising potentials, some limitations still need to be addressed before these nanosystems can transition into clinical settings. There are wide selections of materials and preparation methods of the pH-responsive NPs. Therefore, selecting suitable materials, synthesis methods, and characterization techniques is very important in developing successful pH-responsive NPs. There are several routes to utilize this type of smart NPs to their maximum potential as drug delivery tools. Each route is associated with challenges that need innovative ideas to achieve significant success. It is anticipated that pH-responsive NPs will continue to attract the attention of researchers from different fields such as chemistry, biology, physics, medicine, and nanotechnology to help their progress to clinical applications.

One of the challenges facing the pH-responsive NPs is their low accuracy and off-target delivery due to the heterogeneity of pH across the tumor volume (decreasing from the periphery toward the center of the tumor) and its dependence on the type and stage of cancer. For pH-sensitive NPs relying on reduced pH within endosomes and lysosomes, additional design considerations are needed to ensure that those NPs are internalized via endocytosis, and that appropriate endosomal escape strategies are possible; otherwise, the drugs will be released and degraded by lysosomal enzymes [31]. Another cause of off-target delivery is the reduced pH of lesions and inflammation sites; in such cases, the potential systemic toxicity of pH-responsive NPs can be avoided by using receptor-mediated active targeting. Amongst the different moieties that can be used, monoclonal antibodies (mAbs) and their fragments have garnered a great deal of attention. pH-responsive NPs provide cancer immunotherapy with improved pharmacology and enhanced accumulation of immunotherapeutics in tumor tissues, and reduce off-target side effects [39,80]. For example, Jain et al. [81] developed vascular endothelial growth factor (VEGF) antibody functionalized PEGylated pH-sensitive liposomes loaded with docetaxel (DTX) (VEGF-PEG-pH-Lipo-DTX) for breast cancer therapy. The developed system showed that cellular uptake in MCF-7 cells was increased by 3.17 times compared to free DTX. VEGF-PEG-pH-Lipo-DTX showed a 5.78-fold reduction in IC_50,_ and a 1.70-fold higher apoptotic index compared to free DTX.

Another interesting development is combining pH-sensitivity with other stimuli (i.e., dual/multi-stimuli responsive NPs); NPs responsive to more than one trigger offer an efficient delivery to the targeted sites with highly controlled drug release and reduced systematic toxicity. Nezhadali et al. [82] synthesized pH and temperature-responsive liposomes encapsulating DOX and mitomycin C. The maximum release (98%) from the developed system was obtained at 40 °C and pH 5.5; only 15% was released at 37°C and physiological pH (7.4). In another study, Luo et al. [83] combined NIR phototherapy and chemotherapy to enhance the anti-cancer effects of gold nanoshells coated liposomes. In vivo experiments combining NIR light, pH and temperature as triggers showed the highest antitumor effect with an inhibition rate of 79.65%.

Image-guided drug delivery systems present another promising approach to help overcome the current limitations of cancer drug delivery and therapy strategies. These multifunctional NPs enable the noninvasive assessment of the biodistribution of therapeutic agents, quantification of NPs accumulation at the diseased site, and monitoring of therapeutic efficacy [84]. Several pH-sensitive theranostic systems have been reported for cancer imaging using techniques such as MRI, photoacoustic imaging (PAI), and fluorescence imaging (FI). MRI involves the application of a magnetic field to align the protons in the body with the direction of the applied magnetic field. To obtain an MR signal, energy must be supplied, which, in the case of MRI, is in the form of short radiofrequency (RF) pulses. Application of the RF pulse creates a non-equilibrium state by adding energy to the system; however, once the pulse is switched off, the protons relax back to their equilibrium state, releasing energy that is detected by MRI sensors. There are two relaxation times, namely, spin-lattice (T1) and spin-spin (T2) relaxation. Contrast agents can be used to increase the contrast-to-noise ratio between healthy and diseased tissues [85]. In addition, dual-mode T1/T2 MRI contrast agents have gained much attention because they provide more reliable diagnostic information and higher resolution by the enhanced contrast effects in both T1 and T2 imaging [86]. However, the realization of such contrast agents is challenging because when T1 and T2 contrast agents are combined, they lead to strong magnetic coupling, resulting in undesirable quenching of the magnetic resonance signal. To address this issue, Huang et al. [87] synthesized Mn-porphyrin&Fe_3_O_4_@SiO_2_@PAA-cRGD theranostic nanocomposites. Fluorescent imaging showed that the nanocomposites accumulated in tumor sites by active targeting and were nontoxic to normal cells. Moreover, the nanocomposites exhibited highly sensitive MRI contrast in vivo, accelerating T1 and T2 relaxation to 55 and 37%, respectively.

FI is one of the most commonly used tumor imaging modalities because it offers several advantages, such as ease of operation, and high sensitivity; however, it suffers from low depth penetration and poor signal-to-noise ratio. pH-responsive NPs can improve the signal-to-noise ratio [86]. Qi et al. [88] investigated this concept by developing fluorescent dye (Cy7.5) labeled, pH-responsive copolymer, poly(ethylene glycol)-b-poly(2-(isopropylamino) ethyl methacrylate) (mPEG-b-PDPA-Cy7.5) micelles encapsulating triphenylphosphonium-conjugated pyropheophorbide-a (TPPa, a mitochondria-targeted photosensitizer). The synthesized micellar system was denoted as M-TPPa. The experimental results showed that M-TPPa was quickly endocytosed by cancer cells and immediately dissociated at acidic early endosome to activate fluorescent signals and photoactivity, giving 111- and 151-fold increase in fluorescent signal and singlet oxygen generation (SOG) upon encountering the acidic environment of human HO8910 ovarian cancer cells, respectively.

## 5. Conclusions

Over the past few decades, great progress has been made in developing NPs for drug delivery applications, particularly in cancer therapy. The development of stimuli-responsive NPs has further improved the control of drug release. The diverse materials and methods of preparation allowed pH-responsive NPs to attract more attention compared to the other types of smart NPs. This review presented an overview of promising pH-sensitive molecules and bonds prepared using different materials and preparation methods to be used for cancer therapy. Many reports have shown promising development in preparing successful pH-sensitive NPs; however, these successes remain in the experimental stage and there are still many challenges that need to be overcome (e.g., biocompatibility of some pH-sensitive biomaterials, reproducibility of large-scale production, targeting specificity, and stability) before these systems can reach clinical applications.

## Figures and Tables

**Figure 1 polymers-14-00936-f001:**
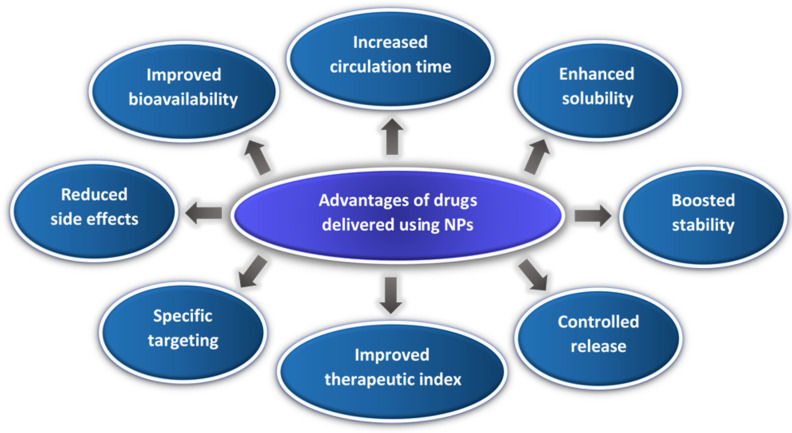
A diagram showing the different advantages of delivering drugs using NPs.

**Figure 2 polymers-14-00936-f002:**
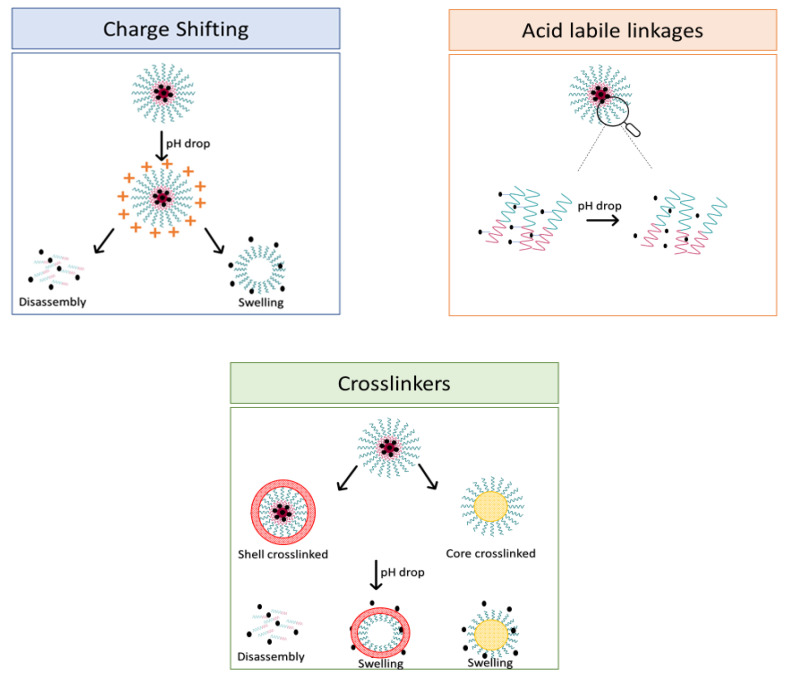
Strategies to design pH-responsive NPs.

**Figure 3 polymers-14-00936-f003:**
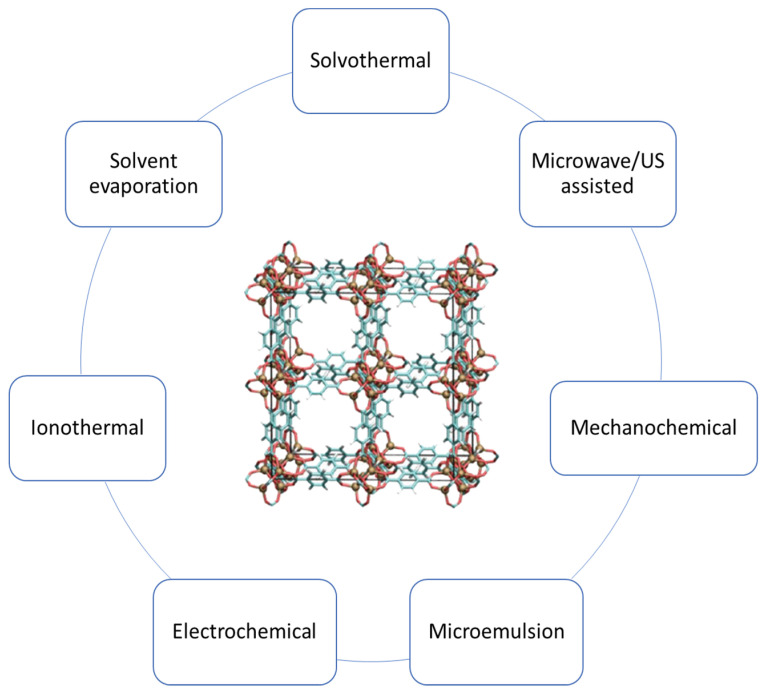
Methods for MOF preparation.

**Figure 4 polymers-14-00936-f004:**
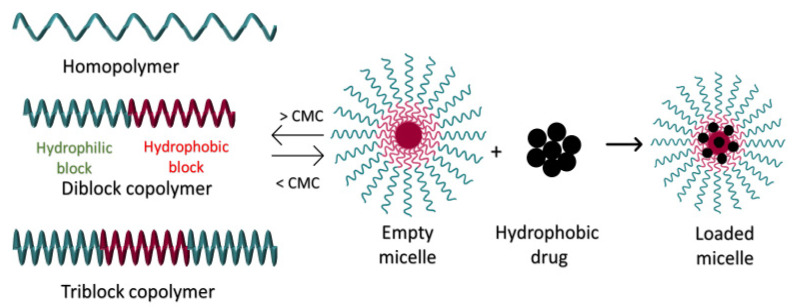
Schematic representation of micelles formation.

**Figure 5 polymers-14-00936-f005:**
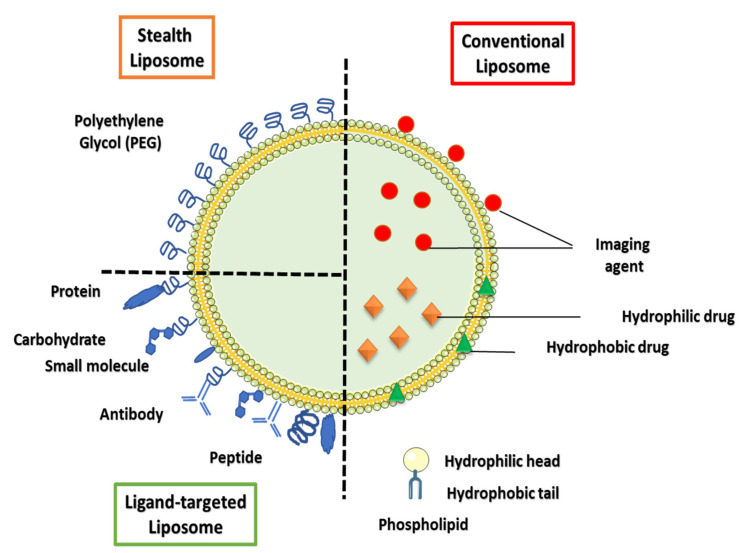
Structure and functionalization of liposomes.

**Table 1 polymers-14-00936-t001:** Advantages and disadvantages of organic NPs [7,9,11,12,13,14,15,16,17].

NPs	Structure	Advantages	Disadvantages
Liposomes	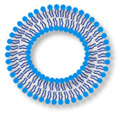	- Biocompatible - Increased circulation time - Amphiphilic - Functional modification - Drug protection - Low toxicity	- May trigger an immune response - Poor stability
Polymeric micelles	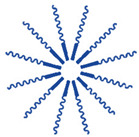	- Biodegradable and biocompatible - Selfassembling - Functional modification - Versatility in chemical composition - Increase solubility of lipophilic drugs - Drug protection	- Occasional cytotoxicity - Degradation of the carrier - Low drug-loading capacity - Difficult to scaleup
Dendrimers	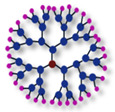	- Uniform shapes - Increased surface area - Increased loading - Can be functionalized with different molecules	- Complex synthesis route - Not used to deliver hydrophilic drugs - High synthesis cost
Solid lipid nanoparticles	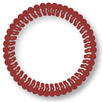	- Soluble and bioavailable - Safe with low toxicity	- Low loading efficiency - Risk of gelation - Drug expulsion due to lipid polymorphism
Nanoemulsions	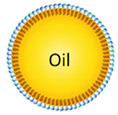	- Stable - Amphiphilic	- Toxicity of surfactants and oils
Hydrogels	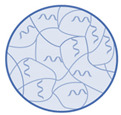	- Ease of administration - Various drug delivery applications, e.g., cell delivery and wound healing	- High water content - Not suitable for hydrophobic drugs

**Table 2 polymers-14-00936-t002:** Advantages and disadvantages of inorganic NPs [7,9,11,12,13,14,15,16,17].

Nanocarrier	Structure	Advantages	Disadvantages
Magnetic nanoparticles	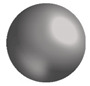	-Uniformity in size-Optical properties enable imaging/theranostic applications	-Potential toxicity-Limited bonding mechanisms
Metal organic frameworks	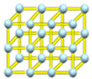	-Large porosity-Large surface area-Open metal sites for reactions	-Low thermal stability-Premature release-Solubility issues under certain conditions
Carbon nanotubes	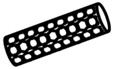	-Multiple functions-Chemical modification-Water dispersible-Biocompatible-Efficient loading	-Potential toxicity-Solubility issues under certain conditions
Quantum dots	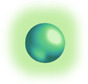	-Beneficial fluorescent properties-Detect, monitor, and deliver drugs to targets	-Induce cytotoxicity
Gold nanoparticles	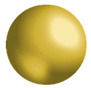	-Increased surface area-Increased loading-Size uniformity-Simultaneous energy delivery	-Potential toxicity

**Table 3 polymers-14-00936-t003:** Advantages and disadvantages of the different stimuli [29,30,31,32].

Type	Advantages	Disadvantages
Visible/near-infrared Light	- High precision - Low cost - Minimum invasiveness - No ionizing radiation	- Low penetration ability (1–10 cm)
pH	- Wide applicability - No need for external triggers	- Low accuracy. - Difficult to maintain their structure - Off-target delivery
Magnetic field	- Imaging/theranostic applications - No limit on tissue penetration - No ionizing radiation	- High cost - Not suitable for tumors located deeper in the body - Possible cytotoxicity
Temperature	- Enhances EPR effect and responsiveness to chemo and radiotherapy - Temperature-sensitive NPs are easy to synthesize - Wide applicability	- Off-target delivery - Internal temperature differences are minimal and highly variable - Stringent demands for NP’s stability
Redox	- High sensitivity	- Off-target delivery - GSHsensitive NPs require association with endosomes
Enzymatic level	- High targeting specificity - Overexpressed in tumors	- Enzyme dysregulation differs between tumors - Limited substrates - Variable expression levels
Ultrasound	- Inexpensive - Not invasive - High safety - Spatiotemporal drug release - No ionizing radiation	- Homogeneous application to large tumors is difficult - Can increase body temperature - Treatment of extensive regions is limited due to cavitation skin burns - Focusing difficulty on organs in motion

**Table 4 polymers-14-00936-t004:** Examples of cationic and anionic polymers and details of their conformational changes in response to the change of pH [38,42].

Polymer Type	Name	Acronym	Structure	Conformational Changes
Anionic	Poly(aspartic acid)	PASP	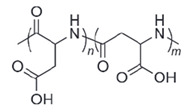	Carboxylate group is deprotonated at pH 7.4 and protonates at pH < 5, which destabilizes the NP.
	Poly(acrylic acid)	PAA	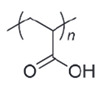	Carboxylate group is protonated at low pH, which destabilizes the NP.
	Poly(2-ethylacrylic acid)	PEAA	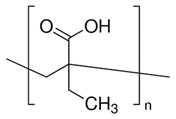	Carboxylate group is deprotonated at pH 7.4 and protonates at pH < 5, which destabilizes the NP.
	Poly(methacrylic acid)	PMAA	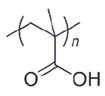	Carboxylate group is deprotonated at pH 7.4 and protonates at pH < 5, which destabilizes the NP.
	Poly-sulfonamides	-	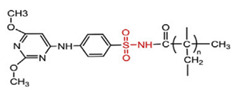	Picks up a positive charge in response to pH decrease, changing the structure of the NP.
Cationic	Poly(b-amino ester)	-	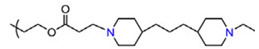	Neutral and hydrophobic at physiological pH, but is ionized and hydrophilic at pH < 6.5.
	Poly(N,N-dimethylaminoethyl methacrylate)	PDMAEMA	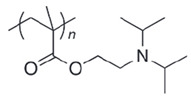	The amine group deprotonates at high pH and protonates/ ionizes at low pH.
	poly(L-histidine)	-	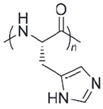	Imidazole ring deprotonates at physiological pH but is protonated at low pH.

**Table 5 polymers-14-00936-t005:** Summary of acid-labile bonds [37,38,45].

Acid-Labile Bond	Structure	Mechanism	Degradation Products
Imine	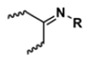	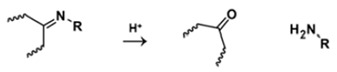	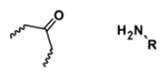
Hydrazone	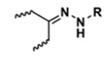	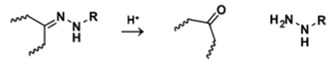	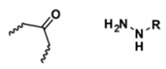
Amides	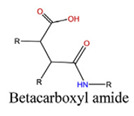	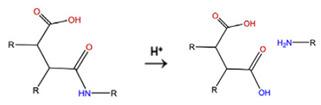	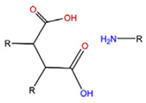
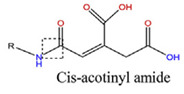	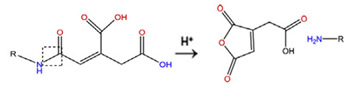	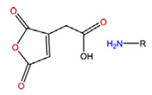
Phenyl vinyl Ether	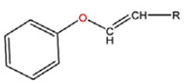	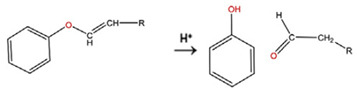	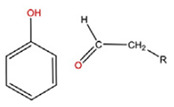
Orthoesters	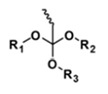	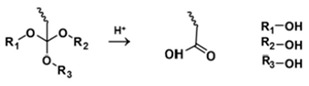	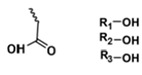
Acetals	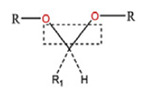	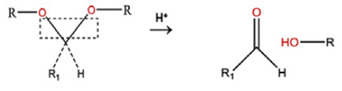	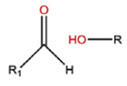
Ketals	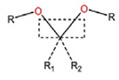	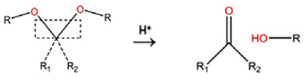	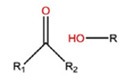
Oxime	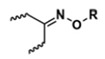	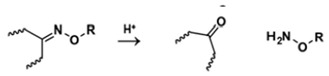	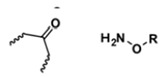

**Table 6 polymers-14-00936-t006:** A summary of studies relevant to pH-triggered micelles in cancer therapy.

Components	Payload	Cancer Cell Line	pH-Triggered Release	Ref.
Poly(ethylene glycol)-b-polycarbonate-b-oligo([R]-3-hydroxybutyrate) (PEG-PKPC-oPHB)	DOX, 8HQ-glu and 8HQ-gla	MCF-7 and HCT-116 cell	- Acidtriggered hydrolysis of ketal groups. - 46% DOX release at pH 7.4 and 77% at pH 5.5.	[59]
Amphiphilic chitosan-g-mPEG/DBA	indocyanine green dye (ICG)	MCF-7	- Cleavage of benzoicimine bonds. - Cumulative release of 23% at pH 5.0 and 9% at pH of 7.4.	[60]
Poly(ethylene glycol)-blockpoly(cyclohexyloxy ethyl glycidyl ether)s	Paclitaxel (PTX) and Nile Red dye	SW620 and DU145 cells	- Cleavage of acetal group.	[61]
Poly (ethylene glycol) methyl ether-b-poly (β-amino esters)	PTX and DOX	A549, MDA-MB-231, A2780 and NCL-H460	- Protonation of tertiary amine residues in PAE block. - Cleavage of cisaconityl linker between copolymer and DOX molecules. - At pH 7.4, the cumulative release of DOX was 9.8%, 75% at pH was 6.5, and 95% at pH 5 at 48 h, respectively.	[62]
1,2-distearoyl-*sn*-glycero-3-phosphoethanolamine-*N*-[methoxy(polyethylene glycol)] conjugated poly(β-amino esters) (DSPE-*b*-PEG-*b*-PAE-*b*-PEG-*b*-DSPE	DOX	B16F10, HepG2 and HeLa cells	- At pH 7.4, the cumulative releases were 15.6%, 27.1% and 30.6% for 2, 24 and 48 h, respectively. - At pH 6.0, the cumulative releases were 28.7%, 56.6% and 61.3% for 2, 24 and 48 h, respectively. - At pH 5, the cumulative releases were 37.5%, 82.3% and 88.9% for 2, 24 and 48 h, respectively.	[63]
Poly(caprolactone) (PCL), poly(ethylene glycol) (PEG), and PCL-bPEG-b-PCL	Pyrene, rohdamine-6G and 5-fluorouracil	-	- The release of 5fluorouracil increased from 13% after 140 h of incubation at pH 7.4 at 37 °C to 52% at pH 5.	[64]

**Table 7 polymers-14-00936-t007:** Polymeric micelles-based drugs for cancer therapy [65].

Product Name	Active Ingredient	Status	Company
Genexol PM	Paclitaxel	Marketed	Samyang, Seongnam, South Korea
NK-911	Doxorubicin	phase II	Nippon Kayaku Co., Tokyo, Japan
NK-105	Paclitaxel	phase II/III	Nippon Kayaku Co., Tokyo, Japan
NC-6004	Cisplatin	phase III	Nanocarrier Co., Chiba, Japan
SP-1049C	Doxorubicin	phase II/III	Supratek Pharma Inc., Quebec, Canada
NC-6300	Epirubicin	phase I/II	Nanocarrier Co., Chiba, Japan

**Table 8 polymers-14-00936-t008:** Commercially available liposomes-based drugs for cancer therapy [68].

Product Name (Year Approved)	Active Agent	Lipid Components	Indication	Company
Doxil^®^ (1995)	Doxorubicin	HSPC, cholesterol; PEG 2000-DSPE	Ovarian, breast cancer, Kaposi’s sarcoma	Sequus Pharmaceuticals, California, USA
DaunoXome^®^ (1996)	Daunorubicin	DSPC and cholesterol	Kaposi’s sarcoma	NeXstar Pharmaceuticals, Colorado, USA
Myocet^®^ (2000)	Mifamurtide	DOPC and POPC	Non-metastatic osteosarcoma	Takeda Pharmaceutical Limited, Tokyo, Japan
Marqibo^®^ (2012)	Vincristine	SM and cholesterol	Acute lymphoblastic leukaemia	Talon Therapeutics, Inc., California, USA
Onivyde™ (2015)	Irinotecan	DSPC, MPEG-2000 and DSPE	Metastatic adenocarcinoma of the pancreas	Merrimack Pharmaceuticals Inc., Massachusetts, USA
Lipoplatin^®^	Cisplatin	SPC-3, cholesterol and mPEG2000-DSPE	Pancreatic adenocarcinoma, NSCLC, HER2/neu negative metastatic breast cancer and advanced gastric cancer	Regulon Inc., California, USA

**Table 9 polymers-14-00936-t009:** A summary of studies relevant to pH-triggered liposomes in cancer therapy.

Lipid Components	pH-Sensitive Component	Payload	Cancer Cell Line	pH-Triggered Release	Ref.
DOPE, CHEMS, DSPE-PEG_2000_	DOPE	DOX	MDA-MB-435S and HeLa cells	- DOPEDVar7lip@DOX release 5times more DOX at pH 5.3 than at pH 7.4.	[72]
Citraconic anhydride (CA), DSPC, DSPE-PEG_2000_	CA	Curcumin	MCF-7 and L929	- Improved release at pH 6.6. - Burst release followed by controlled release.	[73]
DCPA	H_2_O	Ciprofloxacin, red-fluorescent, rhodamine dye	HepG2	- DCPAH_2_O liposomes, accumulated 11times more in the tumor compared to the rest of the body.	[74]
DPPC,DSPE-PEG_2000_, CHOL, DSPE-PEOz_2000_	DSPE-PEOz_2000_	Metformin- and IR780	MDA-MB-231	- pH-responsive drug release helped inhibit mitochondrial respiration.	[75]
HSPC, DSPE-PEG_2000_, C_18_-AI-PEG5000 and C_18_-PEG5000	C_18_-AI-PEG5000 and C_18_-PEG5000	Irinotecan (CPT-11)	MCF-7, BxPC-3 and NIH/3T3	- Release at pH 7.4 was 20%, while at a pH of 6.5, it reached 40%.	[76]
CHEMS, PEG, Nio	pH-sensitive niosomal (Nio) formulation of GTE	Green tea extract (GTE)	MCF-7, HepG2, and HL-60	- Sustained release (77% at pH 5) followed Higuchi release kinetics.	[77]
Egg phosphatidylcholine, CHOL, DSPE-PEG_2000_-angiopep-2	DSPE-PEG_2000_-angiopep-2	Calcium arsenite	HBMEC and C6	- A2–PEG–LP@CaAs released 77.94% at pH 5.5, which is higher than that at pH 7.4 (57.71%) and pH 6.5 (65.32%).	[78]
EPC, PDMAEMA-b-PLMA diblock copolymer	PDMAEMA-b-PLMA	TRAM-34	HEK-293 and GL261	- At pH 7, EPC: PDMAEMAbPLMA 1 released 40% initially then slowly reached up to 55%, EPC:PDMAEMAbPLMA 2 released 30% initially then reached 37% - At pH 5.5, a burst release of 70% for EPC:PDMAEMAbPLMA 1 and 85% for EPC:PDMAEMAbPLMA 2 then reaching almost 100% for both systems.	[79]

## Data Availability

No new data were created or analyzed in this study. Data sharing is not applicable to this article.

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
