# Peer review of "pH-Responsive Nanocarriers in Cancer Therapy"

_polymers, 2022, doi:10.3390/polym14050936_

Round 1
Reviewer 1 Report
The submitted review article is a comprehensive overview on the pH-responsive nanoparticle-based drug delivery systems, which mentions advantages and limitations and the early stage in clinical development. Figures and Tables are informative.
My only comment would be to adapt the title because not only micelles and liposomes are discussed.
Minor
l.55 delete “and”
Error messages at several references.
Author Response
Please find attached the rebuttal letter.

Reviewer 2 Report
Professor Ghaleb A. Husseini group have summarised the pH-responsive Micelles and Liposomes for the treatment of cancer. He has well organised the content information is very useful for the readers. He has listed the various pH-responsive polymers that are very useful in the preparation of a pH-responsive drug delivery system. The overall content is good I strongly recommend to the authors to include one table for commercial products and amphiphilic materials on how to form the micelles and liposomes. The author should include one responsive mechanism paragraph that will be very informative to the readers of this paper. For the above-mentioned comments, Authors must cite some strong articles as follows: Devices for Controlled Release Advancements and Effectiveness. In Design and Development of Affordable Healthcare Technologies (pp. 103-134). IGI Global.
Nanosystems for drug delivery of coenzyme Q10. Environmental chemistry letters, 16(1), 71-77.
Self-assembly of partially alkylated dextran-graft-poly [(2-dimethylamino) ethyl methacrylate] copolymer facilitating hydrophobic/hydrophilic drug delivery and improving conetwork hydrogel properties. Biomacromolecules, 19(4), 1142-1153
Synthesis and multi‐responsive self‐assembly of cationic poly (caprolactone)–poly (ethylene glycol) multiblock copolymers. Chemistry–A European Journal, 23(34), 8166-8170.
Therapeutic Efficacy of Herbal Formulations Through Novel Drug Delivery Systems." In Enhancing the Therapeutic Efficacy of Herbal Formulations, pp. 1-42. IGI Global, 2021
MRI-FI-guided superimposed stimulus-responsive co-assembled liposomes for optimizing transmembrane drug delivery pathways and improving cancer efficacy." Applied Materials Today 26 (2022): 101368.
Liposomes encapsulating artificial cytosol as drug delivery system." Biophysical Chemistry 281 (2022): 106728.
The author should include above-suggested citation and explain in the paragraph the mechanism.
Author Response
Please find the rebuttal letter attached.

Reviewer 3 Report
Paper titled (pH-responsive Micelles and Liposomes in Cancer Therapy) by AlSawaftah et al. is a very interesting article that will be useful for beginner pharmacists who wish to work in pharmaceutical sciences & graphs are very ,meaningful. I recommend the following changes for improvement:
1- Check the abbreviations at the first appearance, some definitions are missing, such as VEGF at page 15.
2- Some references come in package at the end of a paragraph, please use each reference accurately at the site of citation.
Author Response
Please find the rebuttal letter.

Round 2
Reviewer 3 Report
In the revised form of the review article titled (pH-responsive nanocarriers in Cancer Therapy), although it is very hard to check the article for the corrections in presence of the "track change", I guess the authors improved the article according to the reviewer's comments.